# Effect of statin treatment on metabolites, lipids and prostanoids in patients with Statin Associated Muscle Symptoms (SAMS)

Timothy J. Garrett[1]*, Michelle A. Puchowicz[2], Edwards A. Park[3], Qingming Dong[3], Gregory Farage[4], Richard Childress[5], Joy Guingab[1], Claire L. Simpson[6], Saunak Sen[4], Elizabeth C. Brogdon[3], Logan M. Buchanan[3], Rajendra Raghow[3], Marshall B. Elam[3,7]*

1 Southeast Center for Integrated Metabolomics (SECIM), Department of Pathology, Immunology and Laboratory Medicine, University of Florida, Gainesville, Florida, United States of America, 2 Pediatrics-Obesity, University of Tennessee Health Sciences Center, Memphis, Tennessee, United States of America, 3 Department of Pharmacology, University of Tennessee Health Sciences Center, Memphis, Tennessee, United States of America, 4 Department of Preventive Medicine, University of Tennessee Health Sciences Center, Memphis, Tennessee, United States of America, 5 Endocrine Section, Memphis Veteran's Affairs Medical Center, Memphis, Tennessee, United States of America, 6 Department of Genetics, Genomics, and Informatics, University of Tennessee Health Sciences Center, Memphis, Tennessee, United States of America, 7 Cardiology Section, Memphis Veteran's Affairs Medical Center, Memphis, Tennessee, United States of America

* tgarrett@ufl.edu (TJG); melam@uthsc.edu (MBE)

**Data Availability Statement:** The deposited data is publicly available on the website metabolomicsworkbench.com under track ID

## Abstract

### Background

Between 5–10% of patients discontinue statin therapy due to statin-associated adverse reactions, primarily statin associated muscle symptoms (SAMS). The absence of a clear clinical phenotype or of biomarkers poses a challenge for diagnosis and management of SAMS. Similarly, our incomplete understanding of the pathogenesis of SAMS hinders the identification of treatments for SAMS. Metabolomics, the profiling of metabolites in biofluids, cells and tissues is an important tool for biomarker discovery and provides important insight into the origins of symptomatology. In order to better understand the pathophysiology of this common disorder and to identify biomarkers, we undertook comprehensive metabolomic and lipidomic profiling of plasma samples from patients with SAMS who were undergoing statin rechallenge as part of their clinical care.

### Methods and findings

We report our findings in 67 patients, 28 with SAMS (cases) and 39 statin-tolerant controls. SAMS patients were studied during statin rechallenge and statin tolerant controls were stud-ied while on statin. Plasma samples were analyzed using untargeted LC-MS metabolomics and lipidomics to detect differences between cases and controls. Differences in lipid species in plasma were observed between cases and controls. These included higher levels of lino-leic acid containing phospholipids and lower ether lipids and sphingolipids. Reduced levels of acylcarnitines and altered amino acid profile (tryptophan, tyrosine, proline, arginine, and taurine) were observed in cases relative to controls. Pathway analysis identified significant

4462. Batch files for MZmine processing are included in the paper and Supplemental Information.

**Funding:** NIAMS/NIH R21AR0704018 (MBE, RR) National Institutes of Arthritis and Musculoskeletal and Skin Disease (NIAMS) Funder played no role in study design or execution. https://www.niams.nih.gov.

**Competing interests:** The authors have declared that no competing interests exist.

increase of urea cycle metabolites and arginine and proline metabolites among cases along with downregulation of pathways mediating oxidation of branched chain fatty acids, carnitine synthesis, and transfer of acetyl groups into mitochondria.

## Conclusions

The plasma metabolome of patients with SAMS exhibited reduced content of long chain fatty acids and increased levels of linoleic acid (18:2) in phospholipids, altered energy production pathways (β-oxidation, citric acid cycle and urea cycles) as well as reduced levels of carnitine, an essential mediator of mitochondrial energy production. Our findings support the hypothesis that alterations in pro-inflammatory lipids (arachidonic acid pathway) and impaired mitochondrial energy metabolism underlie the muscle symptoms of patients with statin associated muscle symptoms (SAMS).

## Introduction

In the midst of an epidemic of obesity, metabolic syndrome and cardiovascular disease (CVD), nearly 40% of Americans 65-years and older take statins to treat dyslipidemia associated cardiovascular pathology [1, 2]. Due to its safety and efficacy, statin therapy has become the cornerstone of the treatment of atherosclerotic CVD [3]. Unfortunately, statin discontinuation is surprisingly frequent [4, 5] and is associated with a 15–50% higher risk of death from CVD [6–8]. The most common reason patients stop taking these drugs is statin associated myopathy due to statin associated muscle symptoms (SAMS). [9, 10], SAMS patients experience muscle pain (myalgia) stiffness, and weakness with statin treatment, typically without elevation in muscle creatine phosphokinase (CPK) [11]. This disorder is far more common than the rare but life-threatening condition of rhabdomyolysis, high CPK and renal failure [9, 12].

Although the pathogenesis of SAMS is known to be affected by age, gender, ethnicity, low body mass index, renal disease, vitamin D deficiency and concomitant use of other medications [11, 13], none of these factors are uniquely associated with this disorder. The absence of a clear clinical phenotype renders it nearly impossible to prospectively identify patients who are likely to develop SAMS. This is compounded by the absence of biomarkers that are either predictive or diagnostic for SAMS. In the absence of robust biomarkers, physicians have difficulty determining which patients have true SAMS as opposed to ordinary muscle complaints. Similarly, the lack of understanding of the metabolic derangements underlying SAMS has hampered the development of potential mitigating therapies [14, 15].

Examination of molecular pathways in muscle of SAMS patients has provided some insights into the pathogenesis of this disorder. Pathways implicated in the pathogenesis of SAMS include those related to mitochondrial energy metabolism [16] oxidative stress, mitochondrial $Ca^{++}$ efflux, chloride ion channel and lactate efflux in the myocyte [12] and apoptosis [17]. Despite these significant advances, our understanding of the mechanisms underlying these changes remains incomplete.

Metabolomics, the profiling of metabolites in biofluids, cells and tissues, is an important tool for biomarker discovery [18, 19]. Coupled with advanced bioinformatics metabolomic analyses can be used to detect subtle alterations in biological pathways and provide insight into mechanisms underlying disease. Although metabolomics has been used to assess effects of statin treatment in healthy volunteers [20–23], this powerful approach has not yet been applied

to patients with SAMS. In this study we profiled metabolites and lipids using untargeted metabolomic and lipidomic analysis in plasma of patients with SAMS and statin tolerant controls.

## Material and methods

### Study population

Study participants were recruited from the Lipid Metabolism Clinics at the Memphis VA Medical Center. Cases were patients referred to the clinic for evaluation and treatment of SAMS who underwent rechallenge with statin as part of their clinical care. Controls were recruited from the general patient population. The protocol was approved by the Memphis VA Institutional Review Board (IRB) (protocol #1318719). The study was conducted according to the principles expressed in the Declaration of Helsinki. All study participants provided written informed consent prior to initiating study procedures. Study data among participants were collected and managed using REDCap electronic data capture tools hosted at the University of Tennessee Health Science Center, Memphis [24, 25]. REDCap (Research Electronic Data Capture) is a secure, web-based software platform designed to support data capture for research studies, providing 1) an intuitive interface for validated data capture; 2) audit trails for tracking data manipulation and export procedures; 3) automated export procedures for seamless data downloads to common statistical packages; and 4) procedures for data integration and interoperability with external sources.

SAMS patients (Cases) were eligible to participate if they had discontinued the use of statins on two or more occasions due to muscle weakness or myalgia and were willing to undergo rechallenge with statin. The diagnosis of SAMS was made using the National Lipid Association (NLA) definition of probable statin associated muscle symptoms (SAMS) (a myalgia clinical index score of 9–11) [15]. In cases where insufficient information was available to assign a valid NLA score, the Naranjo probability score for adverse drug reactions was used to determine eligibility [26]. Patients were excluded from the study if they had advanced renal disease, active liver disease, advanced cirrhosis, clinically active autoimmune disease, current use of one or more drugs contraindicated for statin use, history of statin associated myonecrosis or rhabdomyolysis or history of unusually severe muscle symptoms with statin therapy. Controls were patients with confirmed absence of statin associated muscle symptoms and who have been compliant with statin therapy for at least one year based on drug dispensing records. Both controls and cases were studied while on statin therapy. Cases underwent rechallenge with either simvastatin, atorvastatin or other statin per lipid clinic protocol. Statin rechallenge continued for up to 4 weeks or until the participant developed myalgia symptoms at which time blood samples were collected. Blood was drawn in the clinic following an overnight (8–12 hour) fast and immediately centrifuged. Plasma samples were flash frozen in liquid nitrogen and stored at -80 degrees Centigrade. Samples were transferred on dry ice to the Center for Integrated Metabolomics at the University of Florida, Gainesville for analysis.

### Participants eligible for analysis

The purpose of the study was to examine differences in the lipidome and metabolome between those with and without SAMS, and how statins alter these profiles in subjects with SAMS. An unanticipated confounding factor in these associations was the effect of fish oil supplements taken by a subset of study participants. The effect of fish oil supplementation on the lipidomic and metabolomic profiles of study participants made detection of changes in metabolites related to case status very difficult using standard data analysis techniques as described above. To compensate for this confounding effect, we excluded 34 participants who were prescribed

fish oil supplements from further metabolomic and lipidomic analyses. The analyses reported herein were derived from the remaining 67 patients comprising 39 controls and 28 cases.

## Metabolomic analyses

**Metabolomics LC-MS analysis.** Metabolites were extracted from plasma samples and LC-MS untargeted metabolomics was performed on a Thermo Q-Exactive Orbitrap mass spectrometer equipped with a Dionex UPLC system (Thermo, San Jose, CA) as described previously [27]. Separation was achieved on an ACE 18-pfp 100 x 2.1 mm, 2 μm column (Mac-Mod Analytical, Inc., Chadsford, PA) with mobile phase A as 0.1% formic acid in water and mobile phase B as acetonitrile (Fisher Scientific, San Jose, CA). The gradient was run flow rate 350 μL/min and consisted of: 0–3 min, 0% B; 3–13 min, 80% B, 13–16 min, 80% B, 16–16.5 min, 0% B. The total run time was 20.5 min. The column temperature was set at 25˚C. Injection volume was 4 μL for negative and 2 μL for positive polarity. All samples were analyzed in positive and negative heated electrospray ionization with a mass resolution of 35,000 at $m/z$ 200 as separate injections. The heated-electrospray conditions are: 350˚C capillary temperature, 3.5 kV capillary voltage, 50 sheath and 10 arbitrary units auxiliary gas. LC-MS injection was done following a sequence of 3 blanks, neat QC, pooled QC, 10 randomized samples, blank, neat QC, pooled QC, 10 randomized samples, and so on.

**Metabolomics data processing.** Percent relative standard deviation of internal standard peak areas were calculated to evaluate extraction and injection reproducibility. The raw files were then converted to mzXML using MS Convert (ProteoWizard, Palo Alto, CA) [28]. MZmine 2 was used to identify features, deisotope, align features and perform gap filling to fill in any features that may have been missed in the first alignment algorithm [29]. The data was searched against SECIM internal retention time metabolite library which was curated with authentic standards. Metabolites identified using an authentic standard are identified with all caps, while those that were searched against a database are in lower case. All adducts and complexes were identified and removed from the data set. The batch files used for MZmine processing are provided as supplemental information (S1 and S2 Files). They are in an XML format that can be opened with MZmine. The full metabolomics dataset can be accessed via https://www.metabolomicsworkbench.org under track ID 4462.

## Lipidomics analyses

**Lipidomics LC-MS analysis.** Lipids were extracted using a modified version of the Folch method as previously described [27]. LC-MS untargeted lipidomics was performed on a Thermo Q-Exactive Orbitrap mass spectrometer with Dionex UHPLC and autosampler. All samples were analyzed in positive and negative heated electrospray ionization with a mass resolution of 35,000 at $m/z$ 200 as separate injections. Separation was achieved on an Acquity BEH C18 1.7 μm, 100 x 2.1 mm column with mobile phase A as 60:40 Acetonitrile:10 mM ammonium formate with 0.1% formic acid in water and mobile phase B as 90:8:2 2-propanol: acetonitrile: 10mM ammonium formate with 0.1% formic acid in water. The flow rate was 500 μL/min with a column temperature of 50˚C. 5 μL was injected for negative ions and 3 μL for positive ions. All samples were analyzed in positive and negative heated electrospray ionization with a mass resolution of 35,000 at $m/z$ 200 as separate injections. The heated-electrospray conditions are: 300˚C capillary temperature, 3.5 kV capillary voltage, 30 sheath and 5 arbitrary units auxiliary gas. LC-MS injection was done following a sequence of 3 blanks, neat QC, pooled QC, 10 randomized samples, blank, neat QC, pooled QC, 10 randomized samples, and so on. Data dependent MS/MS (MS$^2$) and AIF (All Ion Fragmentation) were collected from the pooled samples. A value of '1' or '2' in front of the lipid names signifies that we had

sufficient MS/MS information from $MS^2$ or AIF (respectively) to identify the fatty acyl chains. A value of '3' indicates that we had sufficient MS/MS data to identify only the head group and a value of '4' indicates a search against an in-silico database.

**Lipidomics data processing and analysis.** Percent relative standard deviation of internal standard peak areas were calculated to evaluate extraction and injection reproducibility. Lipidomics data from positive and negative ion modes were processed using LipidMatchFlow software [30]. First, all $MS^2$ raw files were converted to.ms2 and MS raw files to mzXML using MSConvert. A peak list was generated after running MZmine on all mzXML files all.ms2 files and the generated peak lists were used to run LipidMatch to identify features [31].

**Data preprocessing.** Both metabolomic and lipidomic data were blank feature filtered (BFF) to remove features that were not reliably measured. We measured the internal standard reproducibility across all batches and for both metabolomics and lipidomics. The present relative standard deviation (RSD) was less than 30% for all internal standards except the Splash triglyceride internal standard which was less than 40% RSD. Prior to feature selection, the metabolomic and lipidomic datasets underwent a series of preprocessing steps including the following consecutive steps: imputation based on half the minimum value strategy, normalization employing the probabilistic quotient normalization (PQN) method [32], and log-transformation to make data more symmetric and homoscedastic. To address the issue of batch effects, we employed a robust batch effect correction strategy to mitigate potential biases. Utilizing the surrogate variable analysis (sva) R package (version 3.42.0) [33], we implemented the ComBat method [34] to systematically address known batch effects. This approach involved a parametric adjustment, which allowed for the direct removal of any identified batch-related discrepancies, ultimately enhancing the reliability and reproducibility of our findings. The details of our entire computational pipeline for processing can be found at https://github.com/senresearch/SAMSstudypreprocessing. After data preprocessing, statistical analysis was conducted with Metaboanalyst 3.0. Pathway analysis was conducted with metabolite set enrichment analysis (MSEA). The list of metabolites identified as either level 1 or level 3 were separated as either increased or decreased and analyzed for connected pathways using MSEA. Finally, receiver operating characteristic curve (ROC) analysis was conducted to identify biomarkers that could be predictive of SAMS. Principal components analysis (PCA) was used to evaluate overall clustering between blank, cases (CS) and controls (CN) (S1 Fig in S3 File). PCA showed strong clustering of blanks away from CS and CN with minimal batch effects (S1 Fig in S3 File).

## Results

### Characteristics of study participants

Plasma samples were analyzed from a total of 67 patients (28 cases and 39 controls) (Table 1). The mean age of controls was 65.8 years (range 55–80) and of cases was 66 (range 37–84) (P = 0.931). The majority of participants were male in both groups with greater representation of females among cases as compared to controls (P = 0.036) (Table 1). African Americans comprised 38.4% of controls and 18.5% of cases (P = 0.166 for distribution by ethnicity). The average BMI was in the overweight to obese I category but did not differ between groups. The majority of participants were either overweight (BMI 25–30, 37%), Obese I (BMI 30–35, 26%) or Obese II (BMI > 35, 22%). BMI was normal in only 17% of study participants. There was a high prevalence of coronary disease, hypertension, degenerative arthritis and type 2 diabetes among cases and controls (Table 1). The prevalence of hypertension and type 2 diabetes was nominally higher in controls. There was no significant difference between cases and controls with regard to depressive disorder or Post-Traumatic-Stress-Disorder (PTSD). Anxiety

**Table 1. Statin metabolome cohort summary.**

| Characteristic | CONTROLS (N = 39) Mean ± STDEV (Range) Median | SAMS Cases (N = 28) Mean ± STDEV (Range) Median | P = |
|---|---|---|---|
| Age (years) | 65.8 ± 8.5 (42–80) (68) | 66.0 ± 10.7 (37–84) (69.0) | 0.931 |
| **Sex** | | | |
| (Male) | 37 (95.0%) | 22 (78.6%) | 0.036 |
| (Female) | 2 (5.0%) | 6 (21.4%) | |
| **Ethnicity** | | | |
| (Caucasian) | 23 (59.1%) | 22 (88.6%) | 0.166 |
| (African-American) | 15 (38.4%) | 5 (18.5%) | |
| (Native -American) | 1 (2.5%) | 0 | |
| (Asian-American) | 0 | 1 (3.6%) | |
| BMI | 30.5 ± 5.7 (19.1–43.6) (29.7) | 29.8 ± 5.8 (21.6–43.3) (28.6) | 0.725 |
| **Concomitant Diseases** | | | |
| Coronary Artery Disease | 22 (56.4%) | 13 (48.1%) | 0.508 |
| Hypertension | 37 (94.9%) | 21 (77.7%) | 0.036 |
| Type 2 Diabetes | 18 (46.2%) | 6 (28.6%) | 0.047 |
| Congestive Heart Failure | 5 (12.8%) | 0 (0.0%) | 0.053 |
| Degenerative Arthritis | 15 (38.5%) | 9 (33.3%) | 0.670 |
| History of Hypothyroidism | 3(7.7%) | 5 (18.5%) | 0.185 |
| History of Hyperthyroidism | 0 (0%) | 1 (3.7%) | 0.226 |
| Depressive Disorder | 7 (17.9%) | 8 (29.6%) | 0.266 |
| Post-traumatic stress disorder (PTSD) | 5 (12.8%) | 3 (11.1%) | 0.834 |
| Anxiety Disorder | 4 (10.3%) | 8(29.6%) | 0.045 |
| Bipolar Disorder | 0 (0%) | 3 (11.1%) | 0.033 |
| Agent Orange Exposure | 10 (25.6) | 5 (25.5%) | 0.497 |
| History of Rheumatoid Arthritis (currently inactive) | 1 (2.6%) | 2 (7.4%) | 0.353 |
| Spinal Stenosis | 1 (2.6%) | 1 (3.7%) | 0.791 |

* Statistical analysis: JASP release 0.16.4, University of Amsterdam, Department of Psychology and Psychological Methods Unit www.uva.nl (https://jasp-stats.org)

disorder and bipolar disorder were encountered more frequently in cases (P = 0.045 and 0.033 respectively). There was no difference in agent orange exposure.

To identify underlying conditions associated with statin myalgia, the frequency of use of a wide range of medications was determined. The majority showed no difference between cases and controls (S1 Table in S3 File). The use of nonsteroidal anti-inflammatory drugs (NSAIDS), anxiolytics, antidepressants and antipsychotic medications was not significantly different between cases and controls. Not unexpectedly, the use of non-statin cholesterol lowering medications specifically ezetimibe and PCSK9 inhibitors (Alirocumab, Praluent®) was more frequent among cases as was the use of vitamin D supplements (S1 Table in S3 File). SAMS patients had previously been treated with a wide range of statins with rosuvastatin, simvastatin, atorvastatin and pravastatin being most frequent reflecting VA formulary guidelines (S2 Table in S3 File). Most had unsuccessfully tried multiple statins (average number of statins 2.74) (S2 Table in S3 File). Per our lipid clinic protocol, atorvastatin was most frequently selected for rechallenge (40.7%) followed by pitavastatin (33.3%) (S2 Table in S3 File). Predictably, high intensity statins were more frequently prescribed in controls (64.1% vs 7.4%). Statin rechallenge continued until the patient experienced myalgia or up to 30 days if asymptomatic. The mean duration of statin rechallenge was 25.7 days. 66.7% of SAMS cases experienced myalgia with rechallenge (S2 Table in S3 File).

**Table 2. Laboratory characteristics.**

| Characteristic | CONTROLS (N = 39) Mean ± SEM (Range) | SAMS Cases (N = 28) Mean ± SEM (Range) | P = |
|---|---|---|---|
| Creatinine (mg/dl) | 1.14 ± 0.05 (0.7–1.8) | 1.02 ± 0.04 (0.7–2.0) | 0.078 |
| Potassium (mg/dl) | 4.2 ± 0.06 (3.1–4.9) | 4.3 ± 0.10 (3.1–4.9) | 0.385 |
| Calcium (mg/dl) | 9.2 ± 0.25 (8.7–10.4) | 9.5 ± 0.48 (8.5–10.5) | 0.422 |
| CPK (IU/L) | 203 ± 217 (32–1126) | 142.7 ± 142.0 (14–757) | 0.141 |
| CPK (IU/L) (highest) | 192 ± 34 (14–1181) | 129 ± 18 (41–1353) | 0.244 |
| Glucose (mg/dl) | 118 ± 9 (64–295) | 107 ± 5 (79–353) | 0.324 |
| **Total Cholesterol (mg/dl)** | 157.6 ± 5.1 (105–264) | 196.8 ± 9.9 (94–323) | < 0.001 |
| **LDL-C (mg/dl)** | 87.2 ± 4.3 (26–180) | 119.2 ± 8.7 (35–220) | <0.001 |
| HDL-C (mg/dl) | 41.5 ± 1.8 (22–76) | 43.1 ± 2.0 (26–75) | 0.556 |
| Triglyceride (mg/dl) | 144.3 ± 12.1 (60–803) | 172.1 ± 18.6 (71–650) | 0.194 |
| Testosterone (ng/ml) (total) | 324 ± 38 (166–661) | 261 ± 38 (89.2–558.0) | 0.200 |
| Testosterone (ng/ml) (free) | 6.5 ± 0.7 (0.3–12.5) | 5.9 ± 0.7 (2.4–21.8) | 0.552 |
| Vitamin D (ng/ml) | 34.5 ± 2.4 (9.5–70.2) | 34.2 ± 2.2 (16.8–56.4) | 0.919 |
| Antinuclear Antigen (ANA) | 9 (23.1%) | 2 (7.4%) | 0.076 |
| anti-double-stranded DNA | 1/9 (11.0%) | 0/2 (20%) | 0.621 |
| anti-ribonucleoprotein (RNP) | 7/9 (77.8%) | 1/2 (50%) | 0.425 |
| anti-Smith | 1/9 (22.2%) | 0/2 (0%) | 0.621 |
| anti-SSA (Sjogren's) | 2/9 (22.2%) | 0/2 (0%) | 0.461 |
| anti-SSB (Sjogren's) | 0/9 (0%) | 1/2 (50%) | 0.026 |
| Antihistone antibody | 7/23 (30.4%) | 6/23 (26.1%) | 0.543 |
| Histone antibody titer | 2.2 | 1.7 | 0.343 |

## Clinical laboratory measures

Standard clinical analytes (creatine, cpk, glucose, potassium) did not differ between cases and controls nor did HDL-C, triglyceride, testosterone or vitamin D (Table 2). Total and LDL-cholesterol were higher among cases reflecting less frequent use of high intensity statin therapy (Table 2). Antinuclear antigen (ANA) with reflex tests was performed to determine if autoimmune markers were more frequent among cases. Although participants diagnosed with clinically active autoimmune disease were excluded from the study, positive serology (Antinuclear Antigen) was encountered in 7.4% of cases and 23.1% of controls (Table 2). The prevalence of positive serology was similar between cases and controls with the exception of Sjogren's Syndrome-B (SS-B) which was observed more frequently among cases (P = 0.026) (Table 2). There was a relatively high prevalence of positive anti-histone antibody, a marker of drug related autoimmune activity, however the prevalence did not differ between cases and controls (26.1% vs 30.4% respectively, P = 0.343).

## Metabolomic and lipidomic analyses

In this study, lipidomic and metabolomic analyses were conducted in SAMS (statin associated muscle symptoms) patients to discern the metabolic/lipidomic phenotype of SAMS and to better understand the biological basis of SAMS. Samples were analyzed in both positive and negative ionization modes on a Thermo QE high resolution mass spectrometer with reversed-phase chromatographic separation using separate methods for lipids and metabolites and the data were compiled for statistical analysis. We conducted statistical analysis separately for the differing ionization modes as well as the different metabolites and lipid classes. We removed patients who were taking fish oil supplements after finding that this confounded statistical

analyses. The lipidomic analyses identified that linoleic acid containing lipids primarily in phospholipids (PLs) were significantly elevated in SAMS patients while specific triacylglycerol (TG) lipids were decreased. The metabolomic analyses indicated an increase in urea cycle metabolites and arginine/proline metabolism in SAMS patients and a decrease in branched-chain fatty acids.

## Lipidomic analyses

We detected significant differences in plasma lipid and metabolite profiles in SAMS patients undergoing statin rechallenge as compared to statin tolerant control subjects. S3 Table in S3 File shows lipid analysis from the positive ionization mode. The expression was either increased (blue highlight) or decreased (orange highlight) in relation to SAMS. There were at least 51 identified lipids within six lipid categories; the corresponding species are also noted. Increases in phosphatidylcholine (PC)-phospholipids, such as PC(15:0_18:2), were observed (S3 Table in S3 File). Lipids were identified using LipidMatch and those with a 1 indicate identification using $MS^2$ with fatty acyl chain assignment, those with a 2 indicate identification using all ion fragmentation (AIF) with fatty acyl chain assignment, those with a 3 indicated $MS^2$ identification with only head group fragmentation, and those with a 4 indicate a search against the lipid database in LipidMatch. The total number of lipids decreased in SAMS were 33 of which 67% were TG-glycerolipids (S3 Table in S3 File). Also noted were lower levels of acylcarnitines and oxidized lipids, such as OxTG and OxPC in SAMS patients (S3 and S4 Tables in S3 File). These data underscored a trend towards increased expression of phospholipids with 18:2 (linoleic acid) as a fatty acyl chain in SAMS cases. Negative ionization resulted in a detection of 19 significant lipids of which 15 were increased (as expressed by the blue highlight) in relation to SAMS (S4 Table in S3 File). As in positive ionization, there was an increase in PL species in SAMS cases compared to controls with PC dominating the observed species. Again, this increase highlighted a connection to 18:2 containing lipids in SAMS cases.

Fig 1 shows box plots of PC and TG lipids that were consistently and significantly altered in SAMS patients (CS) as compared to controls (CN). The top panel shows results from negative ionization with an increase in 2 PC's containing 18:2 as one of the tails and the bottom panel shows results from positive ionization highlighting an increase in 2 18:2 containing lipids (PC 15:0_18:2, PC 14:0_18:2). Box plots for two TGs that were decreased (TG 17:1_18:1_20:4, TG 18:0_20:2_20:5) and the sulfatide d14:1_20:0 are shown in Fig 1A and 1B respectively.

Table 3 lists all of 18:2 containing lipids which were significantly altered in SAMS patients. Of the 15 lipids selected, 13 were increased and two (TG-glycerolipids) were decreased.

## Metabolomic analyses

Positive ion metabolites that were identified as nominally significant ($p \leq 0.05$) in SAMS patients are shown in S5 Table of S3 File. Data are sorted by direction of change using metabolite pathway analysis to evaluate which metabolic pathways were increased or decreased in SAMS cases. Metabolites in all capitals are level 1 while the others are level 3. Level 1 refers to identification with at least 2 orthogonal data relative to a standard (retention time and $m/z$) while level 3 refers to only $m/z$ accuracy and searched against a database (i.e., HMDB). The $m/z$, retention time (RT) and the ion detected are included. Plasma levels of multiple acylcarnitines were lower in SAMS cases and multiple amino acids were also dysregulated, specifically, tryptophan and tyrosine were higher in SAMS cases while proline, taurine and asparagine were lower (S6 Table in S3 File). Negative ion metabolites that were significantly changed in SAMS cases are depicted in S6 Table of S3 File. As in S5 Table of S3 File, metabolites in all capitals are level 1 while the others are level 3. Concurrent with positive ion mode tyrosine was

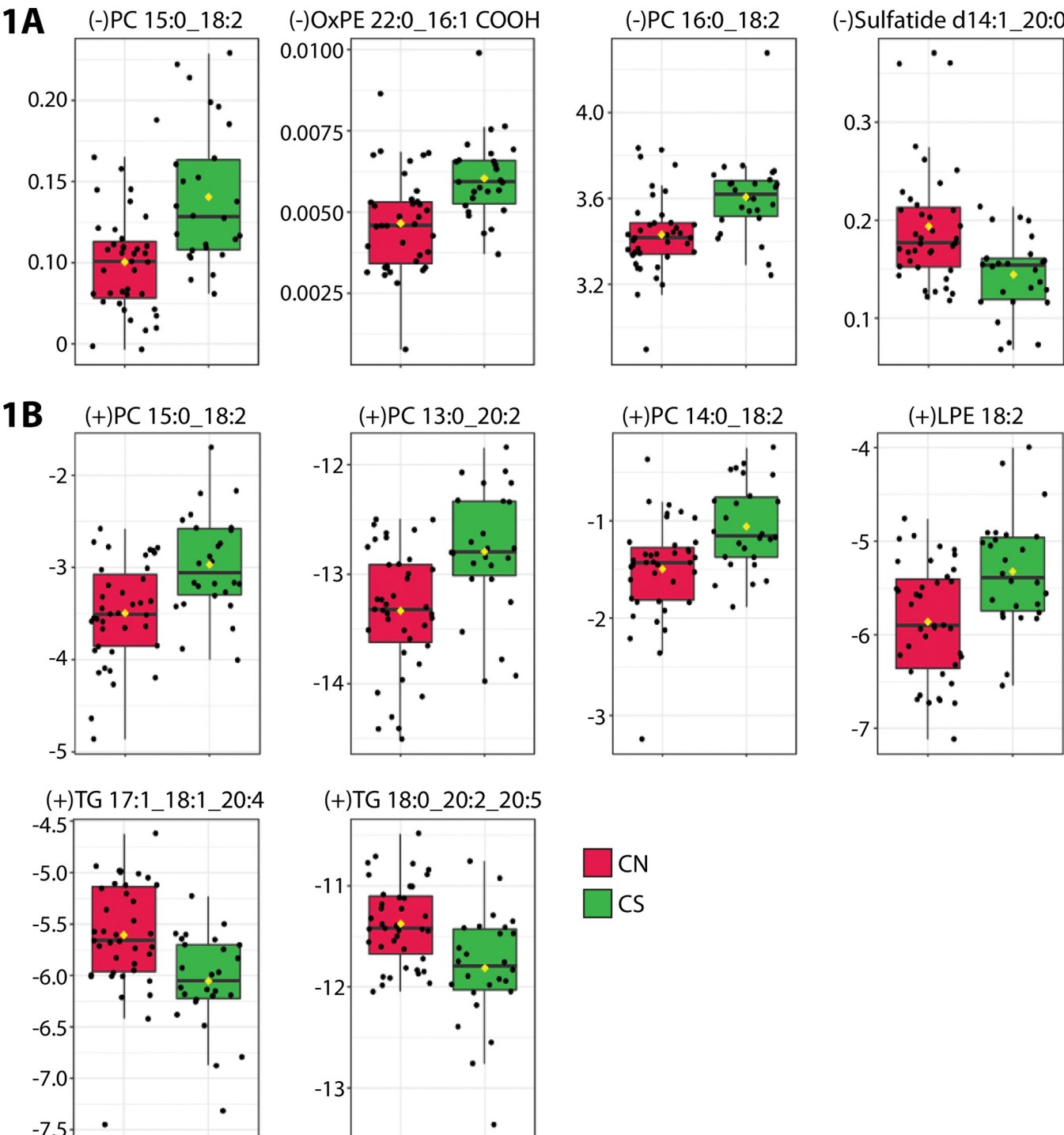

**Fig 1. Phosphatidylcholine (PC) and Triglyceride (TG) lipids altered in SAMS patients (Green box) versus statin tolerant controls (Red boxes).** Top panel. negative ions, bottom panel positive ions.

increased in SAMS cases from the negative ion data and proline, taurine asparagine and N-acetyl-L-serine were decreased. We also observed that the urea cycle intermediates ornithine, citrulline and arginine were increased in cases (Fig 2).

**Table 3. Linoleic acid (18:2, N6) containing lipids were increased (blue highlight) in plasma phospholipids (PC, PI, LPE) of SAMS patients while only 2 glycerolipids (TGs) containing 18:2 were decreased (orange highlight) (TG 16:1_18:2_20:4 and TG 18:2_18:2_20:2).**

| LipidID | Lipid Name | Lipid Class | Lipid Category | t.stat | p.value | negLOG10(p) | FDR |
|---|---|---|---|---|---|---|---|
| posLip64 | 1_LPE(18:2)+H | LPE | Lysophospholipid | -3.4834 | 9.06E-04 | 3.0429 | 0.26634 |
| negLip36 | 1_LPE(18:2)-H | LPE | Lysophospholipid | -3.3121 | 1.54E-03 | 2.8134 | 0.085244 |
| negLip118 | 1_PC(15:0_18:2)+HCO2 | PC | Phospholipid | -4.2712 | 6.69E-05 | 4.1748 | 0.028219 |
| posLip76 | 1_PC(15:0_18:2)+H | PC | Phospholipid | -3.8299 | 2.98E-04 | 3.526 | 0.26634 |
| negLip120 | 1_PC(16:0_18:2)+HCO2 | PC | Phospholipid | -3.6573 | 5.22E-04 | 3.2823 | 0.073437 |
| posLip72 | 1_PC(14:0_18:2)+H | PC | Phospholipid | -3.5789 | 6.71E-04 | 3.1736 | 0.26634 |
| negLip131 | 2_PC(18:0_18:2)+HCO2 | PC | Phospholipid | -3.2535 | 1.83E-03 | 2.7364 | 0.085244 |
| negLip40 | 1_PC(14:0_18:2)+HCO2 | PC | Phospholipid | -3.2458 | 1.88E-03 | 2.7265 | 0.085244 |
| negLip140 | 1_PC(18:2_18:2)+HCO2 | PC | Phospholipid | -3.2214 | 2.02E-03 | 2.6946 | 0.085244 |
| posLip357 | 1_PC(18:2_18:2)+H | PC | Phospholipid | -3.0107 | 3.75E-03 | 2.4263 | 0.32807 |
| negLip60 | 1_PC(18:2_19:0)+HCO2 | PC | Phospholipid | -2.4418 | 1.74E-02 | 1.7586 | 0.32235 |
| negLip75 | 1_PI(18:0_18:2)-H | PI | Phospholipid | -3.1047 | 2.85E-03 | 2.5447 | 0.10032 |
| negLip73 | 1_PI(16:0_18:2)-H | PI | Phospholipid | -2.7807 | 7.14E-03 | 2.146 | 0.17735 |
| posLip1612 | 4_TG(18:2_18:2_20:2)+NH4 | TG | Glycerolipid | 3.0036 | 3.82E-03 | 2.4175 | 0.32807 |
| posLip283 | 1_TG(16:1_18:2_20:4)+NH4 | TG | Glycerolipid | 2.6917 | 9.09E-03 | 2.0413 | 0.32807 |

Interestingly, we observed that the TCA cycle intermediates succinate, malate, and fumarate were increased in SAMS while pyruvate and citrate were decreased consistent with impaired activity of the TCA cycle (S5, S6 Tables in S3 File, Fig 3).

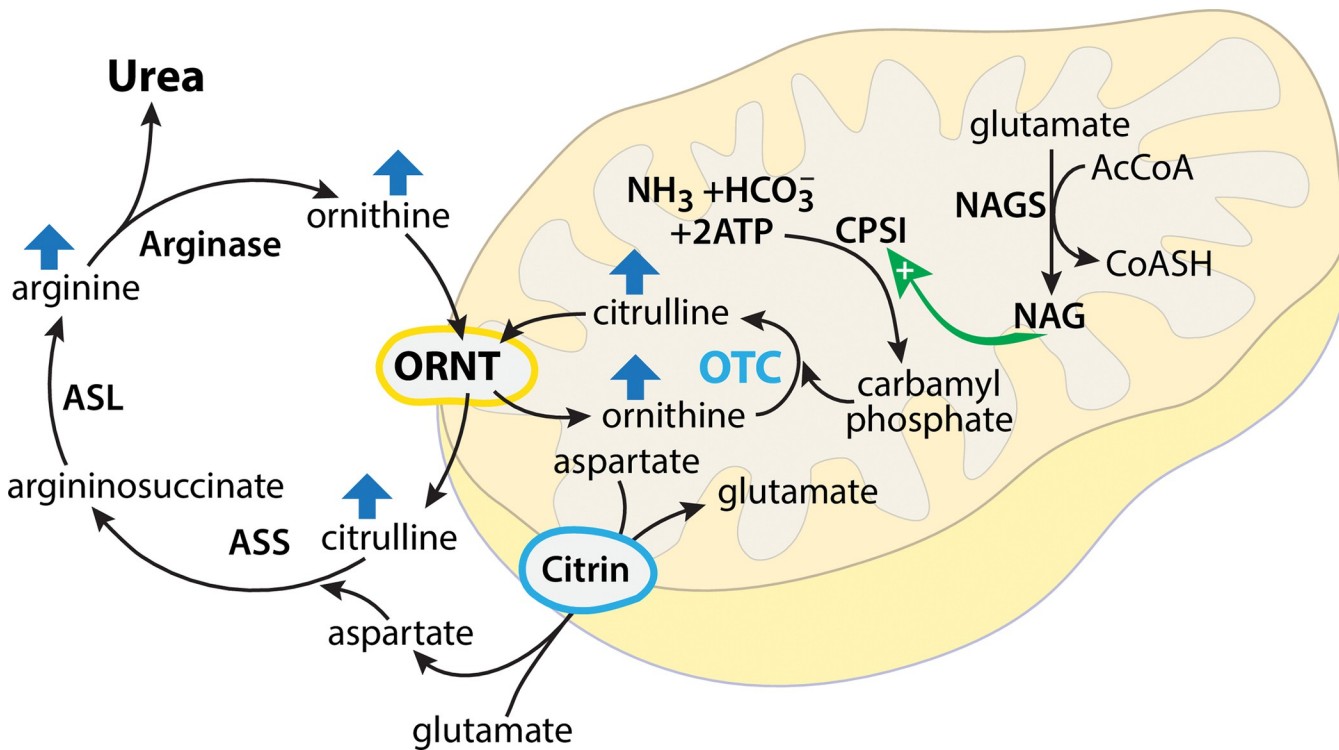

**Fig 2. Increased metabolites of ornithine/citrulline metabolism in SAMS cases are consistent with impaired urea cycle pathway.** Arrows indicate increased expression of specific urea cycle intermediates.

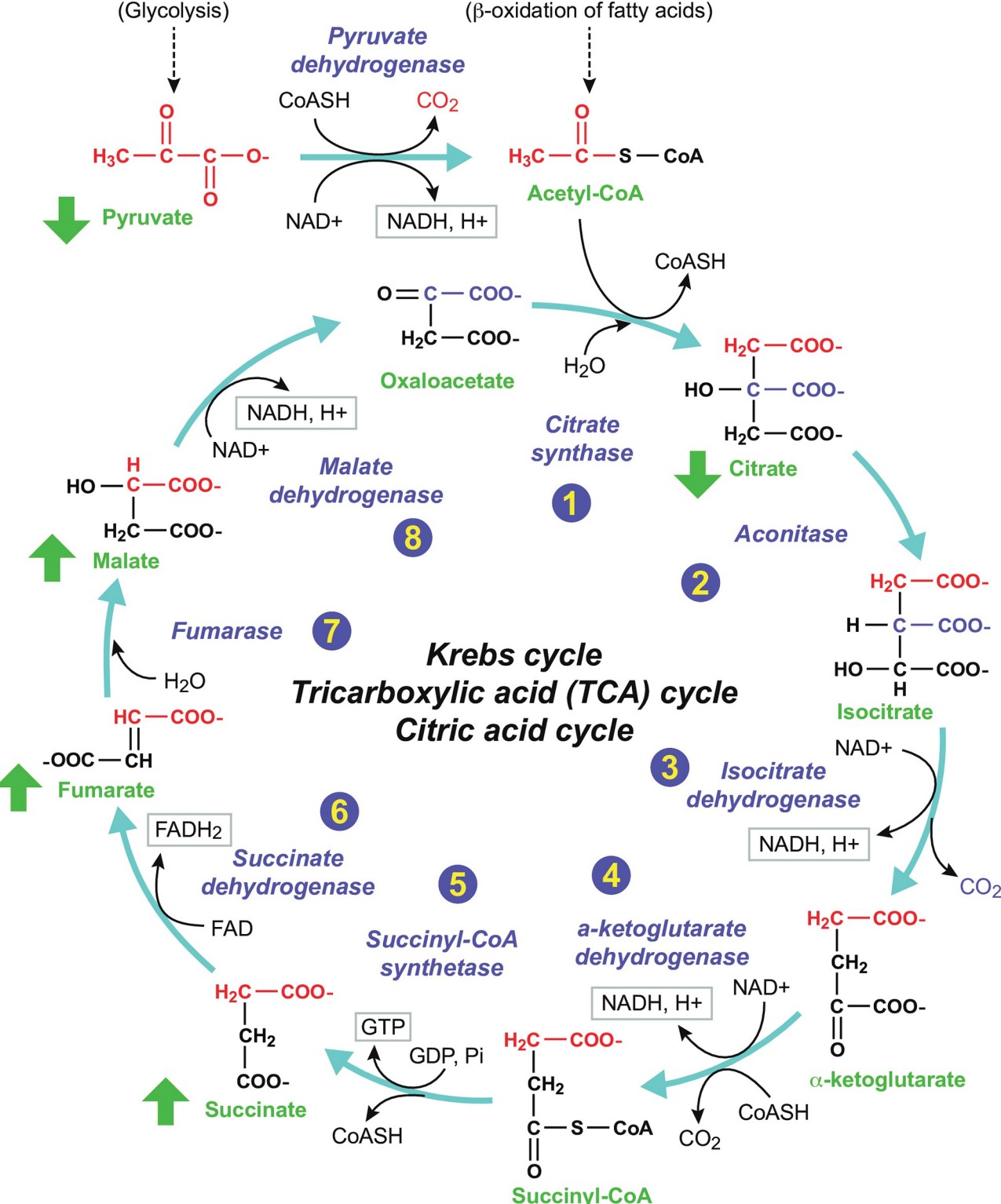

**Fig 3. Altered TCA Cycle intermediates in SAMS patients.** Citric acid cycle intermediates Succinate, Fumarate and Malate were increased whereas pyruvate and citrate were decreased.

Finally, L-carnitine was decreased in cases as were the acylcarnitines propanoylcarnitine, hexanoylcarnitine, butanoylcarnitine, and isovalerylcarnitine, octenoylcarnitine, 3-hydroxy-butyrylcarntine and 2-trans, 4-cis decadienoylcarnitine (S5 Table in S3 File).

## Metabolite set enrichment analysis (MSEA)

Metabolite set enrichment analysis was used to identify the key pathways involved in SAMS (Fig 4). The two most significant pathways that were increased in SAMS cases were the urea

## A. Metabolites Increased in Cases

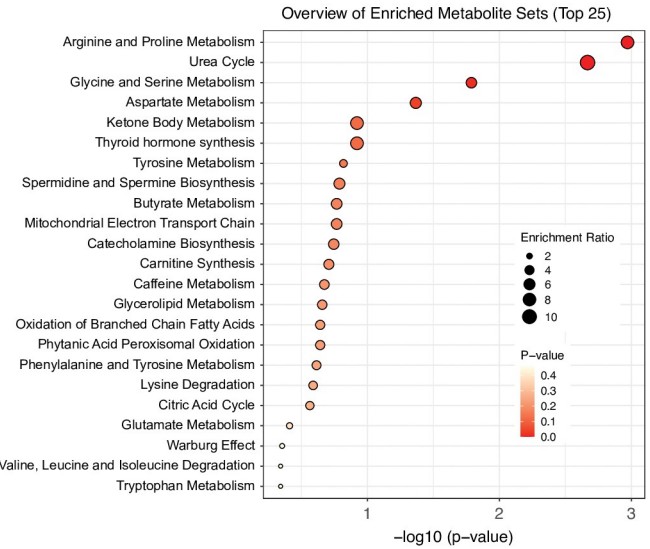

## B. Metabolites Decreased in Cases

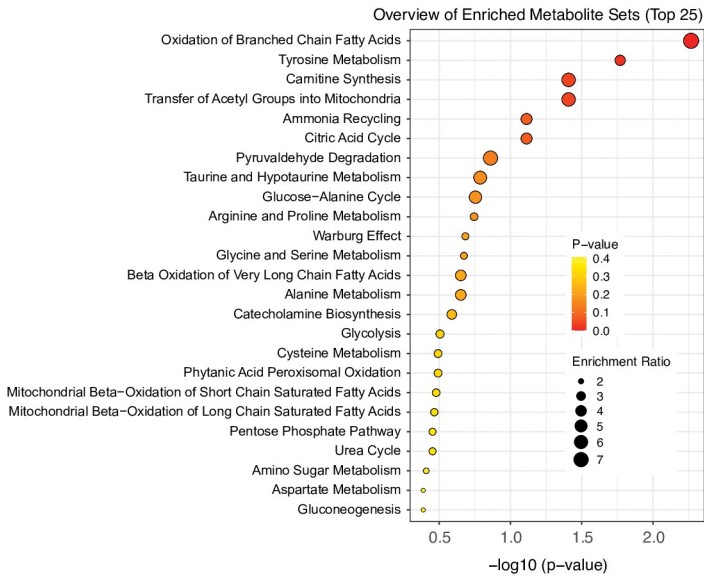

**Fig 4. Metabolite enrichment analysis to identify the key pathways involved in SAMS.** Panel A represents pathways upregulated in SAMS. Panel B represents pathways downregulated in SAMS. Urea cycle and arginine/proline metabolites were increased in SAMS (CS) (A), while oxidation of branched chain fatty acids, carnitine synthesis and acetylation in mitochondria were decreased (B).

cycle and arginine/proline metabolism while the most significant pathways decreased were oxidation of branched chain fatty acids, carnitine synthesis and transfer of acylcarnitine into mitochondria (Pathway maps corresponding to Fig 4 are depicted in S2 Fig of S3 File).

### ROC curve biomarker analysis

To identify potential biomarkers of SAMS we performed Multivariate ROC biomarker analysis with the metabolites and lipids identified as significantly altered in SAMS patients using the nominal $p \leq 0.05$ (Metaboanalyst 3.0). Using support vector machine learning (SVM), we found that a model which included 25 variables had a 0.88 area under the curve (AUC) with a prediction accuracy of 82.2% (Fig 5A and 5B).

Higher features lists of 50 and 100 achieved nearly 90% accuracy; however, we selected the top 25 features to evaluate for potential biomarkers. The selected frequency for the top 25 features (Fig 6A) shows that metabolites were overall the most predictive with 19 of the top 25. Using all 25 features, a cross validation analysis was conducted and 3 CS were identified as CN while 4 CN were identified as cases (Fig 6B). The top 6 features were all metabolites with ribose, pyruvate, glutarate, caproyl glycine, and citrate reduced in cases (CS) while succinate was elevated.

## Discussion

### Summary

Statin related adverse events such as myalgia and myopathy have been well documented, but the metabolic mechanisms associated with statin intolerance remain to be determined [35]. To better define the pathophysiology of SAMS, we performed untargeted metabolomic analyses on plasma samples taken from SAMS patients undergoing rechallenge with statin. The metabolomics profile of SAMS patients following statin rechallenge was compared to that of statin-treated tolerant controls. We found key differences in the plasma metabolome of SAMS

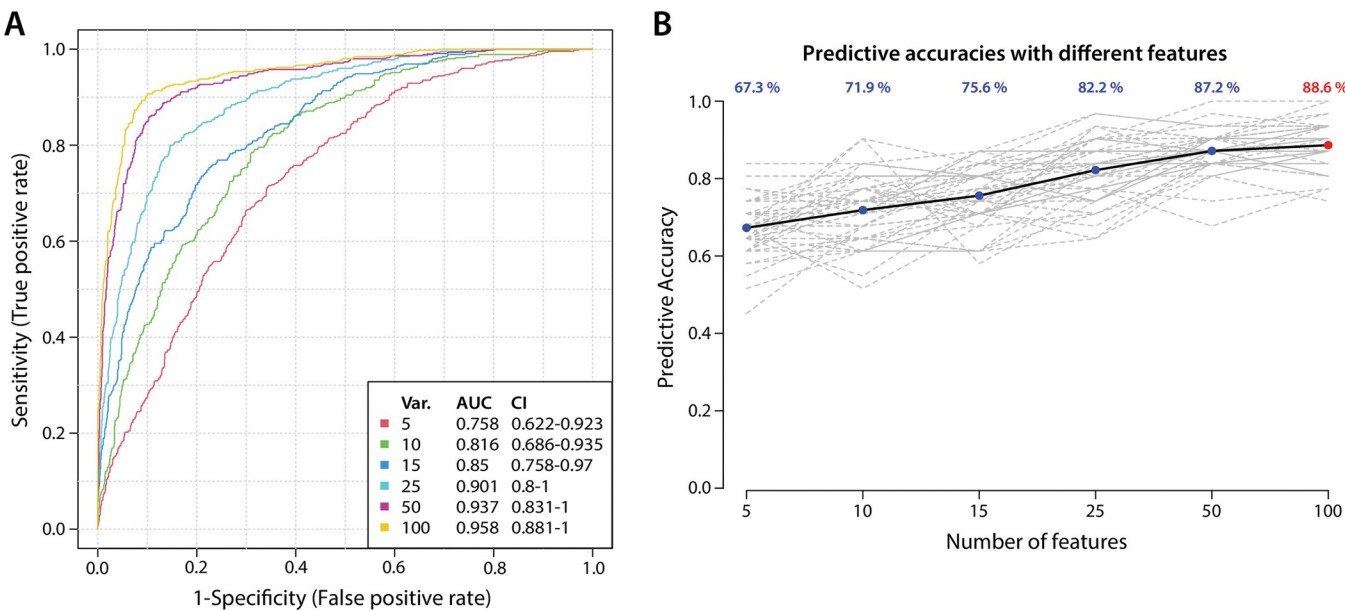

**Fig 5. Receiver operating characteristic curve (ROC) analysis using support vector machine learning (SVM).** Performance of the features from 5 to 100 (A) and the predictive accuracy based on the number of features used in the model (B).

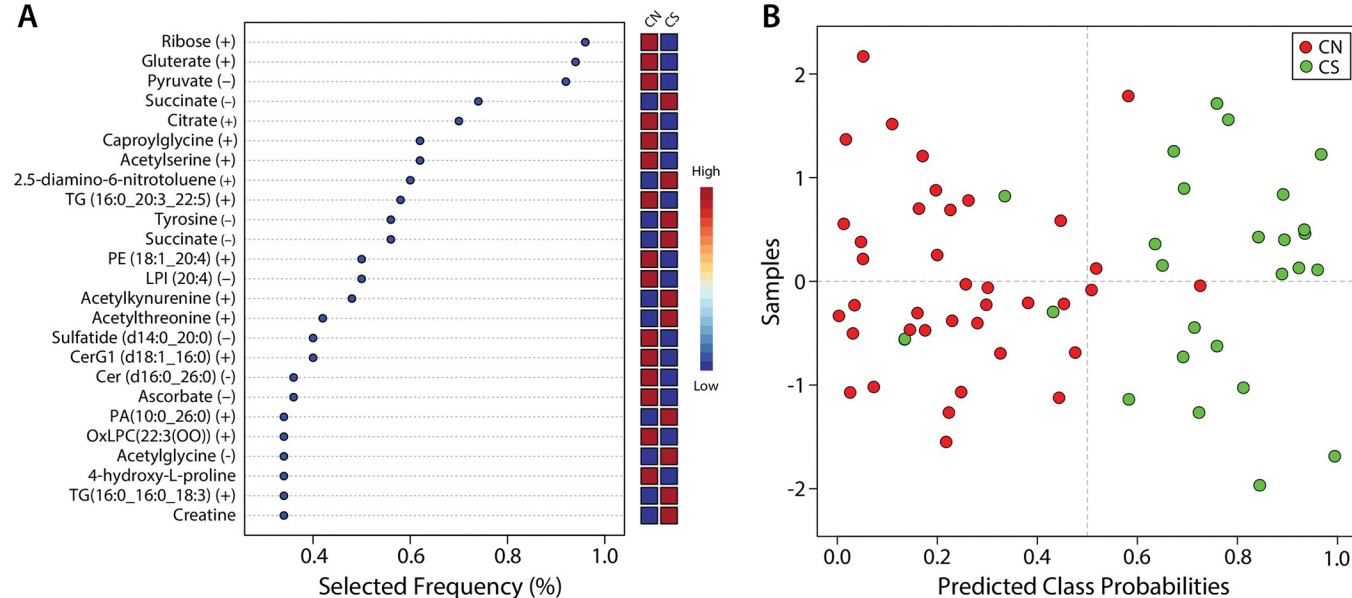

**Fig 6.** The 25 metabolites and lipids which were selected most frequently as predictive (Panel A). Boxes at the right of Fig 4A indicate higher (red) or lower (blue) levels of the analyte. A confusion matrix analysis was conducted, and the classes were predicted well with only 4 controls (CN) predicted as cases (CS) and only 3 CS predicted as CN (Panel B).

patients compared to that of statin tolerant controls including altered fatty acid content of TG and PL, altered mitochondrial energy production pathways and protein turnover (urea cycle). These findings provide novel insights into the pathogenesis of statin associated muscle symptoms (SAMS).

## Significance of observed differences in specific lipids and metabolites

**Phospholipids.** Insofar as linoleic acid (18.2) can be converted to longer-chain fatty acids including the pro-inflammatory eicosanoid arachidonic acid (20:4) [36], the observed increase in phospholipids containing linoleic acid tails in plasma of SAMS patients may be significant. Consistent with this, Rise' et. al. noted increased conversion of linoleic acid to arachidonic acid in simvastatin treated THP-1 cells [37]. Similarly, statin treatment of cultured human hepatocytes (Hep3B) increased levels of lipids containing arachidonic acid (20:4) [38]. Although increased levels of omega-6 polyunsaturated fatty acids including linoleic acid are associated with reduced cardiovascular mortality [39], studies have shown a direct relationship between linoleic acid and elevated inflammatory markers in obesity [40]. Mitochondrial dysfunction has, in turn, been linked to inflammatory response [41].

Furthermore, higher levels of linoleic acid in the diet correspond to a susceptibility to develop chronic pain [42]. While our study is not nutritional, it does suggest that an increase in 18:2 containing lipids in the plasma of SAMS patients might result in increased sensitivity to muscle pain. Our findings suggest that increased availability of linoleic acid underlies a pro-inflammatory state as well as enhanced pain perception in patients with SAMS.

**Urea cycle metabolites.** The urea cycle limits accumulation of ammonia arising from amino acid turnover [43]. Our observation of increased urea cycle intermediates in plasma of SAMS cases as well as elevated levels of the amino acids tyrosine, succinate and tryptophan in SAMS patients is consistent with increased protein turnover. This finding is also consistent with our previous observation of increased expression of genes related to apoptosis and

protein turnover in muscle of SAMS patients [44]. On the other hand formation of citrulline, needed for biosynthesis of nitric oxide and arginine [45] was also increased.

**Metabolites related to mitochondrial energy metabolism.** Our data support a central role of disordered mitochondrial energy metabolism in the pathogenesis of SAMS. Mitochondrial dysfunction has emerged as a key mechanism in the myotoxicity of statins [16, 46] including myopathy due to the inhibition of electron transfer, oxidative stress, induction of apoptosis, and dysregulation of fatty acid oxidation [47]. The effects of statins on mitochondria are linked to depletion of mevalonate pathway intermediates ubiquinone (CoQ10), dolichol and farnesylated proteins [47]. Consistent with impaired mitochondrial metabolism, our observations of altered pools of TCA intermediates suggest an imbalance of the redox state in the cells and mitochondria in cases. In unhealthy cells, as with oxidative stress induced injury, the TCA cycle becomes leaky resulting in an imbalance of metabolite turnover which has been associated with muscle disorders and myopathies [48]. In addition, our observations of down-regulation of metabolites involving oxidation of branched chain fatty acids and mitochondrial fatty acid transport (carnitine) provide evidence of further impact of statin on mitochondrial energy metabolism in SAMS patients.

**Branched chain fatty acids.** Metabolism of branched chain amino acids contributes to energy metabolism and mitochondrial biogenesis [49]. Significantly, supplementation with branched chain amino-acids improved ATP production, muscle strength and cognitive function in a cohort of elderly malnourished patients [50]. The effect of branched chain amino acid supplementation in patients with SAMS has not been reported.

**Carnitine.** Carnitine and its acylated derivatives (acylcarnitines) play an important role in facilitating mitochondrial beta-oxidation of fatty acids by mediating their transport into the mitochondria [51, 52]. Carnitine is a small polar molecule with the highest levels of carnitine occurring in skeletal muscle [52]. Carnitine and acylcarnitine deficiency are associated with hemodialysis, chronic fatigue syndrome and muscular myopathies [52]. The carnitine profile differs among these conditions. In chronic fatigue syndrome, there is a deficiency of acylcarnitines as a result of reduced carnitine palmitoyl transferase activity, whereas in hemodialysis there is a deficiency of carnitine with elevated acylcarnitine due to extensive removal of carnitine in the dialysis procedure [52]. In our analyses, both carnitine and acylcarnitines were reduced. This is consistent with reduced carnitine palmitoyl transferase (CPT-2) activity. CPT-2 deficiency has been associated with statin associated muscle diseases including SAMS [47]. To our knowledge, deficiency of carnitine or acylcarnitines has not been previously reported with statin therapy.

**Ribose.** In our biomarker analysis ribose was the metabolite with the highest prediction frequency. Ribose is an essential substrate for the pentose phosphate pathway which is responsible for adenine nucleotide (AMP) synthesis and recovery of high energy compounds following exercise. Skeletal muscle is particularly dependent upon exogenous ribose for this process [53]. Interestingly, in patients with deficiency of myoadenylate deaminase (MAD) and glycogen phosphorylase (McArdle's disease), ribose supplementation improved exercise tolerance and reduced post-exertional muscle pain or cramps [53]. To our knowledge there are no published reports of the effect of ribose supplementation in patients with SAMS. More work would be needed to determine if ribose therapy for SAMS would be beneficial.

## Comparison of results with those of previous studies

We previously observed altered skeletal muscle gene expression in SAMs patients involving pathways of cellular stress response, apoptosis, protein catabolism, protein prenylation and RAS-GTPase activation [44]. Based on these findings, we posited that myalgia in response to

statins may emanate from cellular stress and resulting post-inflammatory repair and regeneration. We also proposed that these "downstream" effects likely emanated from inhibition of the mevalonate pathway with resulting depletion of critical intermediates (dolichols, ubiquinone, geranylgeranyl phosphate, farnesyl phosphate) needed for protein stability and cell integrity. Although we were not able to detect these mevalonate pathway intermediates in our current analyses, there is considerable evidence that statin-dependent myotoxicity is initiated by attenuated synthesis of intermediates in the mevalonate pathway farnesyl pyrophosphate (FPP) and geranylgeranyl pyrophosphate (GGPP) that mediate protein prenylation and serve as essential lipid anchors for small GTPase proteins [54–56].

The present study is unique in that it applies metabolomics to the study of patients with statin associated muscle symptoms (SAMS) using a rechallenge approach. Several studies in asymptomatic humans and *in vitro* studies have examined the effect of statin therapy itself on the metabolome but not specifically in SAMS patients. The Cholesterol and Pharmacogenetics Study investigators examined metabolomic and lipidomic predictors of statin (simvastatin) response [20–22]. They found that baseline levels of cholesterol, amino acids, fatty acids and plasmalogens were correlated with reductions in LDL-C and CRP following simvastatin therapy. Similarly, Lee et al performed metabolomics analyses on healthy volunteers and patients with hyperlipidemia before and after administration of rosuvastatin [23]. Our findings in statin treated SAMS patients are consistent with their findings of reduced levels of carnitine, diacylglycerol and acylcarnitines and increased levels of fatty acids (FA) lysophophatidylcholines (lysoPC), and phosphatidylcholines (PC) in asymptomatic patients following rosuvastatin treatment [23] and further suggest that these changes are exaggerated in patients with SAMS. In contrast to our findings, in their study of statin treatment in asymptomatic individuals Wurz et. al. observed a reduction in plasma total fatty acids including linoleic acid, no effect of statin therapy on plasma levels of lactate or citrate and a modest increase in pyruvate [57]. Harris et. al. observed decreases in plasma levels of linoleic and linolenic acid and increases in their respective metabolic products, arachidonic acid (AA) and docosahexaenoic acid (DHA) [58]. In this regard, the changes in these analytes observed in our study may be specific to patients with SAMS.

**Biomarker analysis.** Finally, we conducted biomarker analysis to identify metabolites and lipids which could be predictive of SAMS. The TCA cycle was highlighted with pyruvate, succinate and citrate all identified in the top 6 as predictive metabolites. This aligns well with a dysfunction in the mitochondria causing a dysregulation of the TCA cycle. Interestingly, we identified caproyl glycine as significantly downregulated in SAMS and it was also in the top 6 of predictive metabolites. Caproyl glycine, also known as hexanoyl glycine, is in the class of metabolites called acylglycines. Glycine N-acylase is a mitochondrial acyltransferase that transfers fatty acyl groups to glycine from Coenzyme A [59]. The acylglycines hexanoyl glycine, 3-phenylpropionylglycine and suberylglycine are elevated in the urine of patients with medium-chain acyl CoA dehydrogenase deficiency (MCAD) which reduces fatty acid oxidation [60]. Standard of care for identifying metabolic myopathy includes the analysis of acylglycines for any fatty acid oxidation disorders [61]. Overall, the biomarkers we identified relate to a central issue in mitochondrial function.

## Strengths and limitations of the study

There are several strengths to our study including the rigorous selection of cases and comprehensive unbiased evaluation of the plasma metabolome and lipidome. Our unique study of SAMS patients following statin rechallenge allowed direct comparison of plasma metabolome in cases and controls in response to statin treatment. Our use of untargeted metabolomics and

lipidomics allowed a comprehensive assessment of overall metabolic changes associated with SAMS. Our untargeted analysis can be followed up with a targeted metabolomic analysis focusing only on the significantly identified metabolites or lipids, which could be a rapid approach to enable screening more patients and reduce solvent use if HPLC is eliminated (potential for green analysis). Cases were selected from patients referred to our lipid clinic for evaluation of recurrent statin-associated muscle symptoms and the diagnosis of SAMS was confirmed using rigorous diagnostic criteria.

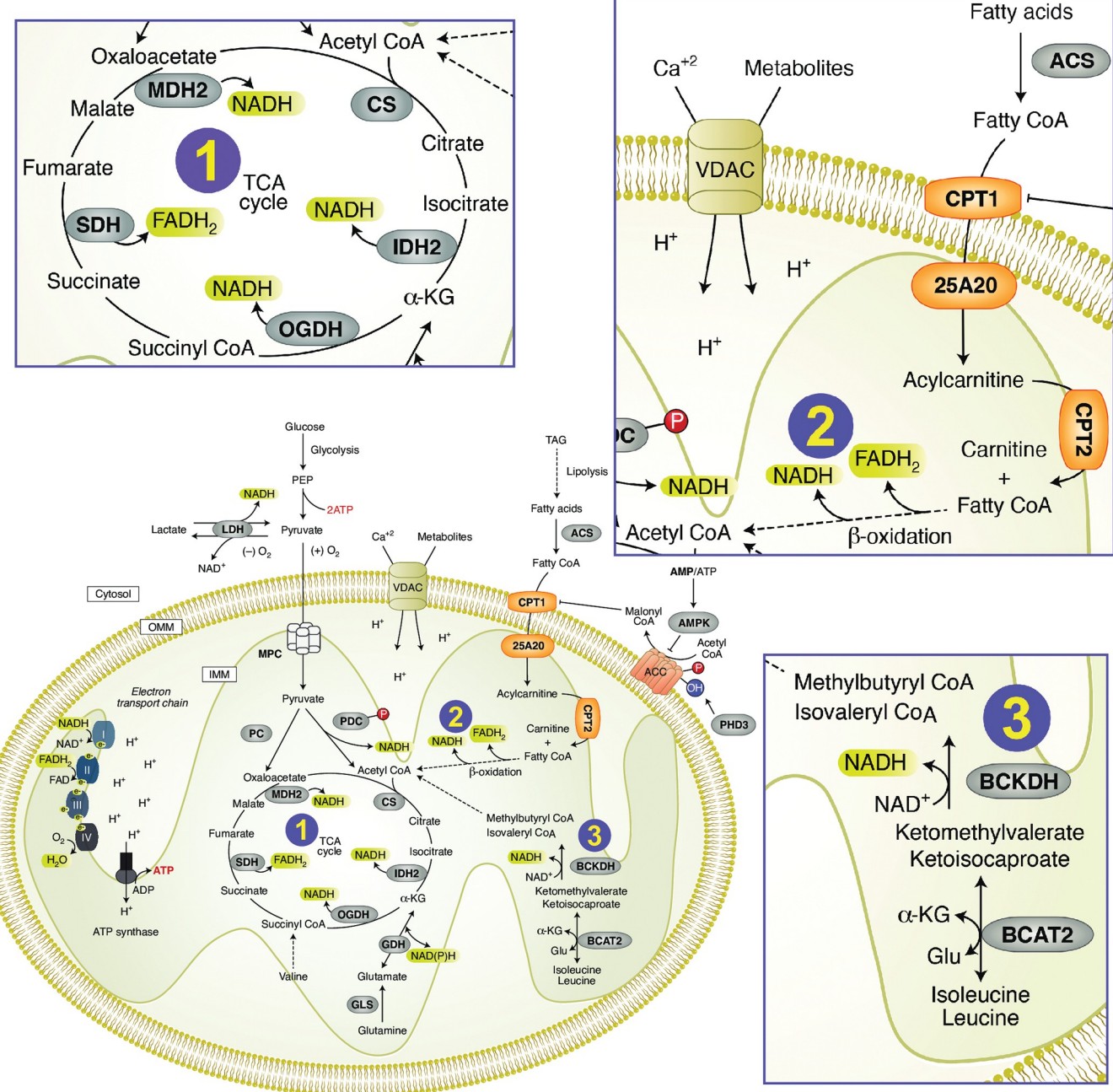

**Fig 7. SAMS patients exhibit defects in multiple mitochondrial energy pathways.** [1] Tricarboxylic Acid (TCA) cycle [2] mitochondrial transport and B-oxidation of fatty acids [3] Oxidation of branched chain fatty acids. Modified from: Spinelli and Haigis Nat. Cell Biol 2018:20:745–54 [64] (reprinted with permission).

Limitations include a relatively modest sample size. Because we studied a highly selected patient population, SAMS patients undergoing rechallenge with statins, the number of subjects studied in this discovery cohort is relatively small. In addition, we had to eliminate patients who were on fish oil therapy after finding it confounded statistical analysis. Second, we analyzed lipids and metabolites in plasma, not muscle. On the other hand, muscle is the largest "organ" accounting for 40% of body weight [62] and serum metabolites have been shown to reproducibly reflect the functional status of muscle in other studies [63]. Third, the diagnosis of SAMS is subjective, there is no specific phenotype or biomarker. In this case, we employed rigorous criteria to confirm the diagnosis of SAMS in study subjects. Fourth, we did not rigidly control dietary intake in study subjects before blood sampling. On the other hand, study subjects were recruited from a homogenous patient population who received uniform dietary counseling as part of their care. Fifth, because relatively few women were studied and African-American's were underrepresented among cases, we were unable to examine differences in metabolites by sex and ethnic group. Finally, although this study has identified potentially important and novel downstream consequences of statin therapy in SAMS patients, identifying the exact mechanism(s) underlying statin myalgia will require targeted mechanistic studies. It is our hope that these findings will serve as a catalyst for the design of such studies.

## Conclusions

The evidence generated by our metabolomic analysis points to a central role of disordered mitochondrial energy metabolism in the genesis of statin associated muscle symptoms. Specifically, we found that multiple pathways involved in generating energy (ATP) are impacted in patients experiencing muscle symptoms of pain, weakness and fatigue associated with statin therapy. As shown in Fig 7, these pathways include [1] the Citric Acid (TCA) cycle, [2] mitochondrial transport and B-oxidation of fatty acids, and [3] oxidation of branched chain amino acids (Fig 7).

We also identified altered metabolism of proteins and amino acids as well as increased levels of the pro-inflammatory fatty acid linoleic acid (18:2) in phospholipids of SAMS patient. Our findings support the hypothesis that alterations in pro-inflammatory lipids (arachidonic acid pathway) and impaired mitochondrial energy metabolism underlie the muscle symptoms of patients with statin associated muscle symptoms (SAMS).

## Supporting information

**S1 File. MzMine batch files for metabolomic processing (negative ion)\*.** *XML format that can be opened with MZmine.
(XML)

**S2 File. MzMine batch files for metabolomic processing (positive ion)\*.** *XML format that can be opened with MZmine.
(XML)

**S3 File.**
(PDF)

## Acknowledgments

We are grateful to all the patients who participated in this study and to Nicholas Townsend, Lonnell Gant, Ashley Armstrong, Lillie Johnson, and Zoe Qualls for assistance in participant recruitment.

## Author Contributions

**Conceptualization:** Richard Childress, Rajendra Raghow, Marshall B. Elam.

**Data curation:** Timothy J. Garrett, Gregory Farage, Joy Guingab, Claire L. Simpson, Saunak Sen, Elizabeth C. Brogdon, Logan M. Buchanan, Marshall B. Elam.

**Formal analysis:** Timothy J. Garrett, Gregory Farage, Joy Guingab, Claire L. Simpson, Saunak Sen, Marshall B. Elam.

**Funding acquisition:** Rajendra Raghow, Marshall B. Elam.

**Investigation:** Timothy J. Garrett, Qingming Dong, Richard Childress, Joy Guingab, Elizabeth C. Brogdon, Logan M. Buchanan, Rajendra Raghow, Marshall B. Elam.

**Methodology:** Timothy J. Garrett, Michelle A. Puchowicz, Edwards A. Park, Qingming Dong, Joy Guingab, Saunak Sen, Elizabeth C. Brogdon, Rajendra Raghow, Marshall B. Elam.

**Project administration:** Richard Childress, Elizabeth C. Brogdon, Logan M. Buchanan, Rajendra Raghow, Marshall B. Elam.

**Supervision:** Edwards A. Park, Saunak Sen, Rajendra Raghow, Marshall B. Elam.

**Validation:** Saunak Sen.

**Writing – original draft:** Timothy J. Garrett, Michelle A. Puchowicz, Rajendra Raghow, Marshall B. Elam.

**Writing – review & editing:** Michelle A. Puchowicz, Edwards A. Park, Qingming Dong, Gregory Farage, Richard Childress, Joy Guingab, Claire L. Simpson, Saunak Sen, Elizabeth C. Brogdon, Logan M. Buchanan, Rajendra Raghow, Marshall B. Elam.

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
