## [Decision Letter · Decision Letter 0]

9 Jan 2023

PONE-D-22-29569Effect of Statin Treatment on Metabolites, Lipids and Prostanoids in Patients with Statin Associated Muscle Symptoms (SAMS)PLOS ONE

Dear Dr. Elam,

Thank you for submitting your manuscript to PLOS ONE. After careful consideration, we feel that it has merit but does not fully meet PLOS ONE’s publication criteria as it currently stands. Therefore, we invite you to submit a revised version of the manuscript that addresses the points raised during the review process.

We look forward to receiving your revised manuscript.

Kind regards,

Anil Bhatia, Ph.D

Academic Editor

PLOS ONE

Journal Requirements:

 "NIAMS/NIH R21AR0704018  (MBE, RR)

National Institutes of Arthritis and Musculoskeletal and Skin Disease (NIAMS)

Funder played no role in study design or execution.

https://www.niams.nih.gov"

   "We are grateful to all the patients who participated in this study and to Nicholas Townsend, Lonnell Gant, Ashley Armstrong, Lillie Johnson, Zoe Qualls for assistance in participant recruitment. This research is supported by grant R21AR0704018 from the National Institute of Arthritis and Musculoskeletal and Skin (NIAMS) of the National Institutes of Health (NIH)."

  "NIAMS/NIH R21AR0704018  (MBE, RR)

National Institutes of Arthritis and Musculoskeletal and Skin Disease (NIAMS)

Funder played no role in study design or execution.

https://www.niams.nih.gov"

   "No authors have competing interests"

Additional Editor Comments:

It was my pleasure to review the PlosOne article, namely “Effect of Statin Treatment on Metabolites, Lipids and Prostanoids in Patients with Statin Associated Muscle Symptoms (SAMS)”.  The overall aritcle points are well described in the report, which is needed to be published. Before going further, I have comments and questions that are necessary to consider before communicating, such as:

Proofreading is needed.In the statement,3 blanks, neat QC, pooled QC, 10 randomized samples, 248 blank, neat QC, pooled QC, 10 randomized samples, and so on. I would suggest adding a data acquisition/sample injection sequence file in the supplementary.On line 185, metabolomics internal standards solution. It is unclear in the article. Specify with catalog numberMy major concern, how was the data normalized before the feature identification, TIC or internal standard. Has the applied 30% CV cutoff in pool QC? All the necessary information about post-data treatment is missing, which is required to publish quality work.Provide the parameters used in mZmine in supplementary.

Reviewers' comments:

Reviewer's Responses to Questions

**Comments to the Author**

1. Is the manuscript technically sound, and do the data support the conclusions?

Reviewer #1: Yes

Reviewer #2: Partly

2. Has the statistical analysis been performed appropriately and rigorously? 

Reviewer #1: Yes

Reviewer #2: No

3. Have the authors made all data underlying the findings in their manuscript fully available?

Reviewer #1: Yes

Reviewer #2: Yes

4. Is the manuscript presented in an intelligible fashion and written in standard English?

Reviewer #1: Yes

Reviewer #2: Yes

5. Review Comments to the Author

Reviewer #1: Garrett et al., conducted a study to understand the effect of statins on patient’s metabolome and lipids associated with statin associated muscle symptoms. This elegant study identified the plasma metabolome of patients with SAMS exhibited reduced content of long chain fatty acids and increased levels of linoleic acid in triglyceride and phospholipid, altered energy production pathways as well as reduced levels of carnitine, an essential mediator of mitochondrial energy production.

1. The details on the blood sample collection were not provided in the materials and methods. Diet significantly influences changes in the metabolome and lipidome of cases and controls. Have the authors collected the fasting blood samples (at least 8-12 hours) to avoid the diet induced changes.

2. Initial analysis of the data by principal component analysis provides the differences between the cases and controls, do the author performed the PCA plots as a part of multivariate data analysis for the metabolomic and lipidomic data?

3. What are the units for the Y-axis values presented in the Figure 1A and 1B? Please provide P-values for each negative and positive ion.

4. Have the authored performed the validation studies to confirm the identity of the identified metabolites and lipids with authentic standards?

5. It is important to perform the validation of the identified metabolites by Receiver Operating Characteristic (ROC) analysis for evaluating the accuracy of a statistical model.

6. All the figure presented in the manuscript are poor and needs high resolution figures.

Reviewer #2: It is recommended to use modeling features and more rigorous analysis to comment on the diagnostic accuracy of the biomarkers. The authors have to not satisfactorily explained the exact mechanism/reason for the increase/decrease seen in lipids or metabolites.

6. PLOS authors have the option to publish the peer review history of their article (what does this mean?). If published, this will include your full peer review and any attached files.

Reviewer #1: No

Reviewer #2: No

---

## [Author Response · Author response to Decision Letter 0]

21 Feb 2023

Please see the "response to reviewers" document.

---

## [Decision Letter · Decision Letter 1]

4 Apr 2023

PONE-D-22-29569R1Effect of Statin Treatment on Metabolites, Lipids and Prostanoids in Patients with Statin Associated Muscle Symptoms (SAMS)PLOS ONE

Dear Dr. Elam,

Thank you for submitting your manuscript to PLOS ONE. After careful consideration, we feel that it has merit but does not fully meet PLOS ONE’s publication criteria as it currently stands. Therefore, we invite you to submit a revised version of the manuscript that addresses the points raised during the review process.

Previously, I raised the flag on the data quality and post processing filters. Such steps are necessary to confirm the data quality and reliability of the features included in the study. I would suggest you to check the following articles or similar to, in which authors described the filters in brief. e.g. Filtering procedures for untargeted LC-MS metabolomics data - PMC (nih.gov) and Five Easy Metrics of Data Quality for LC–MS-Based Global Metabolomics - PMC (nih.gov).

We look forward to receiving your revised manuscript.

Kind regards,

Anil Bhatia, Ph.D

Academic Editor

PLOS ONE

Reviewers' comments:

Reviewer's Responses to Questions

**Comments to the Author**

1. If the authors have adequately addressed your comments raised in a previous round of review and you feel that this manuscript is now acceptable for publication, you may indicate that here to bypass the “Comments to the Author” section, enter your conflict of interest statement in the “Confidential to Editor” section, and submit your "Accept" recommendation.

Reviewer #1: All comments have been addressed

Reviewer #3: (No Response)

Reviewer #4: (No Response)

Reviewer #5: (No Response)

2. Is the manuscript technically sound, and do the data support the conclusions?

Reviewer #1: Yes

Reviewer #3: Yes

Reviewer #4: Yes

Reviewer #5: Yes

3. Has the statistical analysis been performed appropriately and rigorously? 

Reviewer #1: Yes

Reviewer #3: Yes

Reviewer #4: I Don't Know

Reviewer #5: Yes

4. Have the authors made all data underlying the findings in their manuscript fully available?

Reviewer #1: Yes

Reviewer #3: Yes

Reviewer #4: Yes

Reviewer #5: (No Response)

5. Is the manuscript presented in an intelligible fashion and written in standard English?

Reviewer #1: Yes

Reviewer #3: No

Reviewer #4: Yes

Reviewer #5: (No Response)

6. Review Comments to the Author

Reviewer #1: Thanks for taking time to address all queries raised by the reviewers. The manuscript reads much better than the earlier version. Few suggestions or future study designs.

1. When considering human biospecimens for Metabolomics and/or Lipidomic analysis, design your study such that the results are projected in discovery set then validation set of patient and controls. This helps confirmation of the data reproducibility.

2. Although human samples don't show great separation with clustering analysis, a PCA of overall samples including blanks, QC and samples needs to be added in the manuscript to understand the LC-MS analysis QC/QA metrics.

Reviewer #3: The authors have done great job in the manuscript. Experimental design is very clever. Classification and consideration of patients is impressive. Authors have selected individuals very carefully by considering parameters to support the study. They have omitted the patient with important conditions that may hinder their observations.

Authors have done great job in mentioning very basic metabolites and pathways. Their interpretations are awesome. Authors have done great job in mentioning about the strengths and limitations of the study.

Overall impression of an article is very impressive.

There are some revisions that they can focus on to make the manuscript readable :

1. Table 3 second line, 4th column there is a spelling mistake. Carefully, please fix it.

2. Please, make the figures 1A, 1B, 2A and 2B clearer so that reading is easier to read.

3. Figure 4 and Figure 5 are very hard to read. If authors can improve on the clarity of the Figures, reader will enjoy and understand the pathways better.

Reviewer #4: Minor Comments:

1) Line 280: Are African Americans 7.8% or 18.6% of the SAMs cases? Also is the Pvalue=0.166 for % of cases in control vs SAMs for the Caucasian population or African American population as per Table 1?

2) Since the urea metabolic pathway is affected in SAMs cases, it will interesting to see how the values for the control population, who are more diabetic (Table 1), compare to what is known in literature i.e are the basal values of the metabolites related to the urea metabolic pathway elevated in the control cases? That would indicate a higher effect in SAMs cases than what is reported here.

3) The authors do not comment on the BMI of the studied population and its effects on the results.

Major Comment:

1) The effect of gender and ethnicity on the results obtained from metabolic profiles of the studied cases are grossly underrepresented and unattended in the discussion section. The results should be discussed more with respect to the demographics of the population studied here.

2) The figures in the revised document should be revised further as they still are of low resolution with illegible fonts.

General Comment:

This study is more descriptive rather than mechanistic with regards to how statin affects metabolites in SAMs rather than how the effect has downstream consequences. The importance and novelty should be discussed more.

Reviewer #5: (No Response)

7. PLOS authors have the option to publish the peer review history of their article (what does this mean?). If published, this will include your full peer review and any attached files.

Reviewer #1: No

Reviewer #3: No

Reviewer #4: No

Reviewer #5: **Yes: **Pallavi Agarwal

---

## [Author Response · Author response to Decision Letter 1]

25 May 2023

Please see Response to Reviewers Version 3.0 Resubmission Final document. Please note that the original TIF images are degraded in the final PDF file and do not reflect the quality of the original figures.

---

## [Decision Letter · Decision Letter 2]

28 Jun 2023

PONE-D-22-29569R2Effect of Statin Treatment on Metabolites, Lipids and Prostanoids in Patients with Statin Associated Muscle Symptoms (SAMS)PLOS ONE

Dear Dr. ELam,

Thank you for submitting your manuscript to PLOS ONE. After careful consideration, we feel that it has merit but does not fully meet PLOS ONE’s publication criteria as it currently stands. Therefore, we invite you to submit a revised version of the manuscript that addresses the points raised during the review process.

We look forward to receiving your revised manuscript.

Kind regards,

Anil Bhatia, Ph.D

Academic Editor

PLOS ONE

Journal Requirements:

Reviewers' comments:

Reviewer's Responses to Questions

**Comments to the Author**

1. If the authors have adequately addressed your comments raised in a previous round of review and you feel that this manuscript is now acceptable for publication, you may indicate that here to bypass the “Comments to the Author” section, enter your conflict of interest statement in the “Confidential to Editor” section, and submit your "Accept" recommendation.

Reviewer #1: All comments have been addressed

Reviewer #3: All comments have been addressed

Reviewer #4: (No Response)

2. Is the manuscript technically sound, and do the data support the conclusions?

Reviewer #1: Yes

Reviewer #3: Yes

Reviewer #4: Yes

3. Has the statistical analysis been performed appropriately and rigorously? 

Reviewer #1: Yes

Reviewer #3: Yes

Reviewer #4: Yes

4. Have the authors made all data underlying the findings in their manuscript fully available?

Reviewer #1: Yes

Reviewer #3: Yes

Reviewer #4: Yes

5. Is the manuscript presented in an intelligible fashion and written in standard English?

Reviewer #1: Yes

Reviewer #3: Yes

Reviewer #4: Yes

6. Review Comments to the Author

Reviewer #1: (No Response)

Reviewer #3: The authors have improved the manuscript much better than it's initial submission. The content is very informative.

Reviewer #4: The figures 1 and 2a are pixelated to the extent that they are illegible especially Figure 2a. The quality of figures can be improved.

7. PLOS authors have the option to publish the peer review history of their article (what does this mean?). If published, this will include your full peer review and any attached files.

Reviewer #1: No

Reviewer #3: No

Reviewer #4: No

---

## [Author Response · Author response to Decision Letter 2]

7 Jul 2023

Academic Editor Comments:

I have personally reviewed the references and confirmed all as active listings in PubMed with the exception of reference #35 (Patterson et al). This is a methodologic paper authored by Dr. Garrett that describes procedures used in our analyses. The journal Metabolomics is not cited in PubMed however the article is readily accessible on the Metabolomics Journal website. I have included a link to this article at the end of the References section (line 1032 of the manuscript) in order to provide the reader with access. I confirm that no manuscripts cited have been retracted or contain Errata.

Reviewer #3 

The authors have improved the manuscript much better than the initial submission. The content is very informative

We thank the reviewer for this comment.

Reviewer #4. 

The figures 1 and 2a are pixelated to the extent that they are illegible especially Figure 2a. The quality of figures can be improved.

We agree with the reviewer that the version of these figures provided to reviewers in the PDF document generated by the online submission system are degraded. Fortunately, the original TIF files uploaded to the Editorial manager system are of good quality. In response to the reviewers’ comments all figures have been redrawn by a graphic artist and are greatly improved. As part of the submission process these figures have been uploaded to the PACE system and have passed quality control.

---

## [Decision Letter · Decision Letter 3]

30 Aug 2023

PONE-D-22-29569R3Effect of Statin Treatment on Metabolites, Lipids and Prostanoids in Patients with Statin Associated Muscle Symptoms (SAMS)PLOS ONE

Dear Dr. Elam,

Thank you for submitting your manuscript to PLOS ONE. After careful consideration, we feel that it has merit but does not fully meet PLOS ONE’s publication criteria as it currently stands. Therefore, we invite you to submit a revised version of the manuscript that addresses the points raised during the review process.

We look forward to receiving your revised manuscript.

Kind regards,

Anil Bhatia, Ph.D

Academic Editor

PLOS ONE

Journal Requirements:

Reviewers' comments:

Reviewer's Responses to Questions

**Comments to the Author**

1. If the authors have adequately addressed your comments raised in a previous round of review and you feel that this manuscript is now acceptable for publication, you may indicate that here to bypass the “Comments to the Author” section, enter your conflict of interest statement in the “Confidential to Editor” section, and submit your "Accept" recommendation.

Reviewer #4: (No Response)

Reviewer #6: (No Response)

Reviewer #7: (No Response)

2. Is the manuscript technically sound, and do the data support the conclusions?

Reviewer #4: Yes

Reviewer #6: Yes

Reviewer #7: Partly

3. Has the statistical analysis been performed appropriately and rigorously? 

Reviewer #4: Yes

Reviewer #6: Yes

Reviewer #7: Yes

4. Have the authors made all data underlying the findings in their manuscript fully available?

Reviewer #4: Yes

Reviewer #6: Yes

Reviewer #7: Yes

5. Is the manuscript presented in an intelligible fashion and written in standard English?

Reviewer #4: Yes

Reviewer #6: Yes

Reviewer #7: No

6. Review Comments to the Author

Reviewer #4: Figure 2 is illegible especially the small fonts in Panel B. Please replace this figure with one at a higher resolution. The rest of the comments were addressed.

Reviewer #6: This manuscript is well written and is based on good empirical evidence and therefore makes an original contribution. Only few minor changes are required before it can be published:

a) The introduction part can be reframed by addressing the hypothesis and findings in the last paragraph of the introduction and the para related to metabolomics can be included before the proposed outline of the study.

b) In Materials and methods, the authors can cite the references for Metabolomics LC-MS analysis and Lipidomics Sample Extraction.

c) The gap between two lines needs to be checked and rectified wherever necessary throughout the manuscript.

d) In table 2, the value of P= needs to be mentioned in the first row.

Reviewer #7: Thank you for an opportunity to review your manuscript, which describes findings of your investigation, which tries to describe metabolomics profile of patients who suffer from Statin Associated Muscle Symptoms. It is paramount goal and authors should be congratulated for their effort, choice of methodology and an identification of certain pathways occurring in patients with Statin Associated Muscle Symptoms.

Our main suggestions/ comments are mainly referring to presentation of your results and discussion.

While it is possible to follow your results for someone who is familiar with concept of lipidomics/metabolomics it is almost impossible to understand the concept for reader of PLUS ONE who never heard before about metabolomics.

Along the same lines, you manuscript is way too long, which additionally makes it even more difficult to understand.

I would suggest following changes:

1. Please shorten your introduction, in general it should have not be longer than 1 page

2. Please avoid repeating your results in discussion except of first paragraph

3. Please rewrite your discussion trying to follow proposed scheme

a. Main results of your investigation

b. How do your results compare to similar studies if they exist

c. Clinical significance of your results

d. Limitations of your study

e. Conclusions

In current manuscript your conclusions took well over one page.

Additionally, I believe it would be beneficial for reader to understand 2 stages of usual metabolomics/lipidomics analysis. First step would be untargeted analysis and second (confirming step) targeted, which ideally would include green analysis skipping HPLC step.

In order for better understanding of your hypothesis/finding you should include simple schemes presenting an altered metabolic pathways occurring in patients with Statin Associated Muscle Symptoms. Some of your figures try to present it, however they are still difficult to read for someone who is not biochemist.

In summary, this is very ambitious project, which provides metabolomic insight into pathophysiology of Statin Associated Muscle Symptoms but presentation of results

and subsequent pathophysiologic explanation requires significant improvement and clarity.

7. PLOS authors have the option to publish the peer review history of their article (what does this mean?). If published, this will include your full peer review and any attached files.

Reviewer #4: No

Reviewer #6: No

Reviewer #7: No

---

## [Author Response · Author response to Decision Letter 3]

18 Oct 2023

Dear Dr. Bhatia,

Thank you for the invitation to further revise and resubmit our manuscript “Effect of Statin Treatment on Metabolites, Lipids and Prostanoids in Patients with Statin Associated Muscle Symptoms (SAMS). We also thank you and the reviewers for helpful and insightful comments. Version 4.0 of the revised manuscript shows great additional improvement thanks to these recommendations. In this “Response to Reviewers” we respond to the additional points raised by yourself and the reviewers as outlined below. 

We sincerely hope that with these additional revisions you will find this manuscript acceptable for publication in PLOS ONE.

Marshall B. Elam PhD MD

Professor, Pharmacology and Medicine

Reviewer #4: Figure 2 is illegible especially the small fonts in Panel B. Please replace this figure with one at a higher resolution. The rest of the comments were addressed.

Response: To increase font size and improve legibility figure 2 now presents the tabular results. The pathway maps are now presented as Supplemental figure S8.

Reviewer #6: This manuscript is well written and is based on good empirical evidence and therefore makes an original contribution. Only few minor changes are required before it can be published:

a) The introduction part can be reframed by addressing the hypothesis and findings in the last paragraph of the introduction and the para related to metabolomics can be included before the proposed outline of the study.

Response: We agree with this approach and have modified and shortened the introduction accordingly.

b) In Materials and methods, the authors can cite the references for Metabolomics LC-MS analysis and Lipidomics Sample Extraction.

Response: We have shortened and consolidated the sample extraction and LC-MS analysis paragraphs by citing the reference for metabolite and lipid extraction. Thank you for this suggestion. 

c) The gap between two lines needs to be checked and rectified wherever necessary throughout the manuscript.

Response: We have removed any gaps between lines.

d) In table 2, the value of P= needs to be mentioned in the first row.

Response: I believe the reviewer is referring to the header that identifies the values in the fourth column as being P values.

Reviewer #7: Thank you for an opportunity to review your manuscript, which describes findings of your investigation, which tries to describe metabolomics profile of patients who suffer from Statin Associated Muscle Symptoms. It is paramount goal and authors should be congratulated for their effort, choice of methodology and an identification of certain pathways occurring in patients with Statin Associated Muscle Symptoms.

Our main suggestions/ comments are mainly referring to presentation of your results and discussion.

While it is possible to follow your results for someone who is familiar with concept of lipidomics/metabolomics it is almost impossible to understand the concept for reader of PLUS ONE who never heard before about metabolomics.

Along the same lines, you manuscript is way too long, which additionally makes it even more difficult to understand.

I would suggest following changes:

1. Please shorten your introduction, in general it should have not be longer than 1 page

Response: We have significantly shortened the introduction.

2. Please avoid repeating your results in discussion except of first paragraph

3. Please rewrite your discussion trying to follow proposed scheme

a. Main results of your investigation

b. How do your results compare to similar studies if they exist

c. Clinical significance of your results

d. Limitations of your study

e. Conclusions

Response: Thank you for pointing this out, based on your recommendations we have revised the discussion to be more cohesive and concise. 

In current manuscript your conclusions took well over one page.

Response: The conclusions section has been modified and is now less than one page.

Additionally, I believe it would be beneficial for reader to understand 2 stages of usual metabolomics/lipidomics analysis. First step would be untargeted analysis and second (confirming step) targeted, which ideally would include green analysis skipping HPLC step.

Response: Thank you for this comment. We added a sentence in the strengths and limitations section to address the targeted approach. 

The following text was added: “Our untargeted analysis can be followed up with a targeted metabolomic analysis focusing only on the significantly identified metabolites or lipids, which could be a rapid approach to enable screening more patients and reduce solvent use if HPLC is eliminated (potential for green analysis).”

In order for better understanding of your hypothesis/finding you should include simple schemes presenting an altered metabolic pathways occurring in patients with Statin Associated Muscle Symptoms. Some of your figures try to present it, however they are still difficult to read for someone who is not biochemist.

We agree that the affected pathways are complex. To improve comprehension for non-biochemists we have taken the following steps: 1. We have streamlined and simplified the discussion of the involved pathways (TCA Cycle, Urea Cycle, and pathways of Mitochondrial Energy Metabolism). 2. Explanatory figures for the Urea and TCA cycle are now viewed in the context of the corresponding sections of the Results. 3. Figure 8 (now figure 7) summarizing the effects of statin on the interrelated pathways of mitochondrial energy metabolism has been modified with the addition of “exploded” views of the three affected pathways. 

In summary, this is very ambitious project, which provides metabolomic insight into pathophysiology of Statin Associated Muscle Symptoms but presentation of results

and subsequent pathophysiologic explanation requires significant improvement and clarity.

We thank the reviewers for their useful and insightful comments and guidance.

---

## [Decision Letter · Decision Letter 4]

3 Nov 2023

Effect of Statin Treatment on Metabolites, Lipids and Prostanoids in Patients with Statin Associated Muscle Symptoms (SAMS)

PONE-D-22-29569R4

Dear Dr. Elam,

We’re pleased to inform you that your manuscript has been judged scientifically suitable for publication and will be formally accepted for publication once it meets all outstanding technical requirements.

Kind regards,

Anil Bhatia, Ph.D

Academic Editor

PLOS ONE

Additional Editor Comments (optional):

Reviewers' comments:

Reviewer's Responses to Questions

**Comments to the Author**

1. If the authors have adequately addressed your comments raised in a previous round of review and you feel that this manuscript is now acceptable for publication, you may indicate that here to bypass the “Comments to the Author” section, enter your conflict of interest statement in the “Confidential to Editor” section, and submit your "Accept" recommendation.

Reviewer #4: All comments have been addressed

Reviewer #6: All comments have been addressed

2. Is the manuscript technically sound, and do the data support the conclusions?

Reviewer #4: Yes

Reviewer #6: Yes

3. Has the statistical analysis been performed appropriately and rigorously? 

Reviewer #4: Yes

Reviewer #6: Yes

4. Have the authors made all data underlying the findings in their manuscript fully available?

Reviewer #4: Yes

Reviewer #6: Yes

5. Is the manuscript presented in an intelligible fashion and written in standard English?

Reviewer #4: Yes

Reviewer #6: Yes

6. Review Comments to the Author

Reviewer #4: (No Response)

Reviewer #6: (No Response)

7. PLOS authors have the option to publish the peer review history of their article (what does this mean?). If published, this will include your full peer review and any attached files.

Reviewer #4: No

Reviewer #6: No

---

## [Editor Report · Acceptance letter]

7 Dec 2023

PONE-D-22-29569R4 

Effect of Statin Treatment on Metabolites, Lipids and Prostanoids in Patients with Statin Associated Muscle Symptoms (SAMS) 

Dear Dr. Elam:

I'm pleased to inform you that your manuscript has been deemed suitable for publication in PLOS ONE. Congratulations! Your manuscript is now with our production department. 

Kind regards, 

on behalf of

Dr. Anil Bhatia 

Academic Editor

PLOS ONE